# Real-space observation of ferroelectrically induced magnetic spin crystal in SrRuO₃

S. D. Seddon [1], D. E. Dogaru[1], S. J. R. Holt [1], D. Rusu[1], J. J. P. Peters[1,2], A. M. Sanchez [1] & M. Alexe [1✉]

Unusual features in the Hall Resistivity of thin film systems are frequently associated with whirling spin textures such as Skyrmions. A host of recent investigations of Hall Hysteresis loops in SrRuO₃ heterostructures have provided conflicting evidence for different causes for such features. We have constructed an SrRuO₃-PbTiO₃ (Ferromagnetic – Ferroelectric) bilayer that exhibits features in the Hall Hysteresis previously attributed to a Topological Hall Effect, and Skyrmions. Here we show field dependent Magnetic Force Microscopy measurements throughout the key fields where the 'THE' presents, revealing the emergence to two periodic, chiral spin textures. The zero-field cycloidal phase, which then transforms into a 'double-q' incommensurate spin crystal appears over the appearance of the 'Topological-like' Hall effect region, and develop into a ferromagnetic switching regime as the sample reaches saturation, and the 'Topological-like' response diminishes. Scanning Tunnelling Electron Microscopy and Density Functional Theory is used to observe and analyse surface inversion symmetry breaking and confirm the role of an interfacial Dzyaloshinskii–Moriya interaction at the heart of the system.

[1] Department of Physics, University of Warwick, Coventry, UK. [2] School of Physics, Trinity College Dublin, Dublin, Ireland. ✉email: m.alexe@warwick.ac.uk

The emergence of a topological Hall effect (THE)[1,2] in SrRuO₃ (SRO) revealed by unusual peaks in Hall resistivity has been attributed to presence of a skyrmion phase. Such skyrmion signature features have been previously reported in SRO thin films when combined with a spin-orbit coupling layer SrIrO₃[3,4] and ferroelectric systems[5,6]. Both systems were constructed to induce a Dzyaloshinskii–Moriya interaction (DMI)[3,6–8], since the asymmetric exchange interaction is key in skyrmion phase stabilization[9]. Later reports have contradicted this hypothesis assuming a two-stage switching process in ultrathin, bare SRO films due to unit cell differences in the film thickness at substrate step edges[10,11]. A non-chiral solution for the THE was identified for SRO/SrIrO₃ structures[12,13], rejecting the presence of skyrmions. An approach to solve this controversy is to investigate the real-space magnetic structure hunting for the fingerprint of a skyrmion lattice. We have investigated the magnetic domain evolution in ferroelectric-SRO system, where dielectric polarisation is expected to induce a strong symmetry breaking, identifying a chiral spin crystal that emerges over the THE magnetic field range. Resembling a skyrmion lattice and governed by the same magnetic interactions, this chiral spin crystal has been theoretically predicted to cause a THE similar to skyrmions.

SRO is an itinerant ferromagnet with a bulk $T_c$ of ~150 K[14] frequently employed as a bottom contact in epitaxial thin film heterostructures due to its agreeable lattice match with common substrates[15,16]. Despite extensive literature attention exploring transport properties[3,4,6,7,10–12,17], much remains to be understood of this system. Direct observation of skyrmions at the nanoscale is experimentally challenging and their existence is often inferred from transport measurements, more precisely from the appearance of an unusual peak in the Hall resistivity. Magnetic force microscopy (MFM), as a real-space imaging technique, has been successfully employed previously to provide convincing evidence of skyrmions[18–20]; however, there is no compelling evidence that such chiral magnetic structures exists in SRO-ferroelectric bilayer films.

The DMI is an antisymmetric electron exchange interaction responsible for the stabilisation of Skyrmions, arising from broken inversion symmetry in either the bulk or at the surface of a material. As mentioned, the interfacial DMI (iDMI) was shown to be at the heart of thin film systems that exhibited a THE-like response[3,6–8]. Such an interaction relies on broken surface inversion symmetry, which is expected when combining metallic perovskite magnetic materials with a ferroelectric[5,6,21], giving rise to Rashba spin-orbit coupling[22], which in turn acts as the dominant contributing factor to the iDMI.

We perform here in-depth studies of the magnetic domain pattern in the field range where the topological Hall peak occurs. We reveal by MFM measurements emergent periodic chiral domains well correlated with the abnormal peak in the Hall response usually assigned to THE[23,24]. We show theoretically and with atomic-resolution scanning transmission electron microscopy (STEM) that the system exhibits broken surface inversion symmetry, itself causing a large iDMI. Here, we report the real-space observation of a chiral spin crystal previously identified in other materials but never visualised in real space[25–28]. We also demonstrate that ferroelectric polarisation induces such a magnetic chiral phase in a ferromagnet–ferroelectric bilayer system.

## Results

We investigate in this study SRO/PbTiO₃ bilayers epitaxially grown on vicinal SrTiO₃(100) substrates. We selected PbTiO₃ as potential candidate to induce symmetry breaking in SRO as previous works showed that ferroelectric BaTiO₃[6] and BiFeO₃[5] induces THE. Previous work demonstrated that Pb-based ferroelectrics generate a

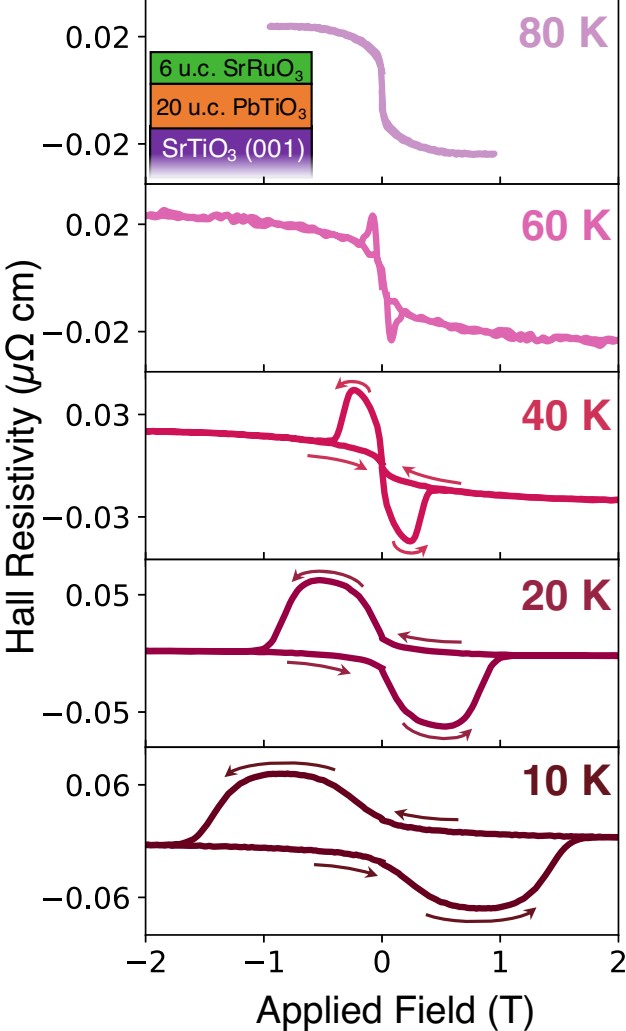

**Fig. 1 Temperature evolution of Hall measurements.** Emergence of a topological-like Hall response below 80 K in SRO–PTO–STO heterostructures. The linear Hall contribution was subtracted. Field-sweep directions are indicated by the arrows. Inset shows the schematic of the system.

stronger interface effect potentially due to a much larger polarisation value[29]. A schematic of the system is provided in Fig. 1, where u.c. refers to unit cells.

The as-grown system shows emergent topological-like Hall effect below 80 K that is characterized by two symmetric peaks in the field-dependence of the transversal resistivity (Fig. 1). The width and the magnitude of the peaks decrease with increasing temperature, a trend expected as the coercive field of the ferromagnetic layer also increases with temperature[14]. The contributions from an anomalous Hall effect are minimal. The sign of the anomalous Hall effect, taken as the sign of the Hall resistivity saturation value at positive fields, transitions from positive to negative at around 20 K, whereas the hysteresis is drastically diminished above 40 K. Therefore, the THE is the main contribution in the Hall response below the high-field saturation value (both forward and backwards field directions).

The magnetic domains at 40 K in the field range where the topological Hall peak occurs have been analysed in depth here. This specific temperature or 40 K was selected for these measurements as the THE peak width allowed for a comprehensive study over the whole peak field range with sufficiently small field increments

between MFM images and therefore efficiently scanning the evolution of the domains. The MFM measurements protocol is as follows. The sample was first field polarised with −2 T and a MFM image was acquired. The MFM images acquired under saturation field show some usual small areas of topographic interference, potentially from dust particles, however it is mostly featureless—void of any phase shifts indicative of magnetic domains. No sign of step terraces is evident. Figure 2b presents MFM scans acquired in 0.02 T increments between 0 and 0.48 T, the range of the THE at 40 K. The scans in Fig. 2b were selected to display key domains that form throughout this region. Fast Fourier transforms (FFTs) are also provided as they reveal the evolution of periodicity and are consequently quantified in Fig. 3.

Weak stripe domains appear at zero field (Fig. 2b) indicating domain pattern formation. Thus, magnetic moments in the absence of an applied field revert to a domain pattern that minimises the energy cost of the system, rather than remaining in a saturated state, as is expected of bare SRO systems until the application of a coercive field. In this magnetic ground state, the average width of the alternating magnetic stripes is ~70 nm, as opposed to the step terraces seen in previous topographic measurements (see Fig. S1 of Supplementary Information), found to be on average ~200 nm, so whilst the inclination of the miscut may dictate the stripe propagation direction, they are not directly correlated with step edges. The lack of any step terraces in high-field images adds confidence to these as magnetic features, and

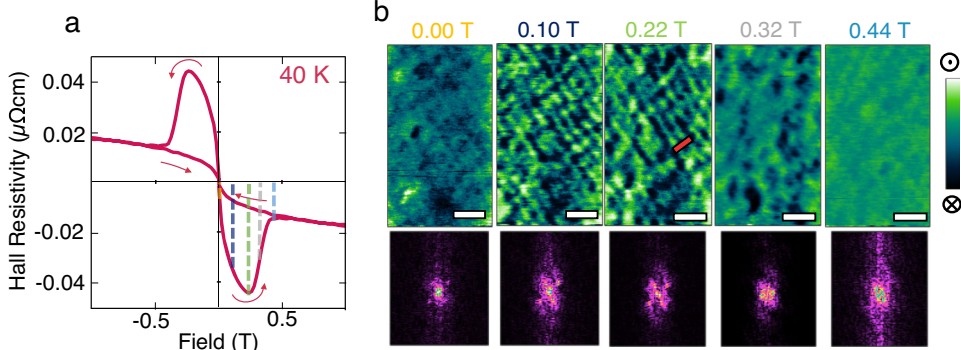

**Fig. 2 MFM experimental data correlated with Resistive Hall data. a** Resistive Hall curve acquired at 40 K. **b** Sequential MFM images acquired at 0.00, 0.10, 0.22, 0.32, and 0.44 T. Each scan is colour coded with dashed lines in **a** to correlate with the Hall measurements. Scale bars are 500 nm, colour bars are −0.3° to 0.3°. Below each image is a fast Fourier transform (FFT) of the respective image. All MFM images and their corresponding FFT are available in the Supplementary Information as well as in Supplementary Movie 1. Red bar at 0.22 T indicates location of line scan used in Fig. 3f.

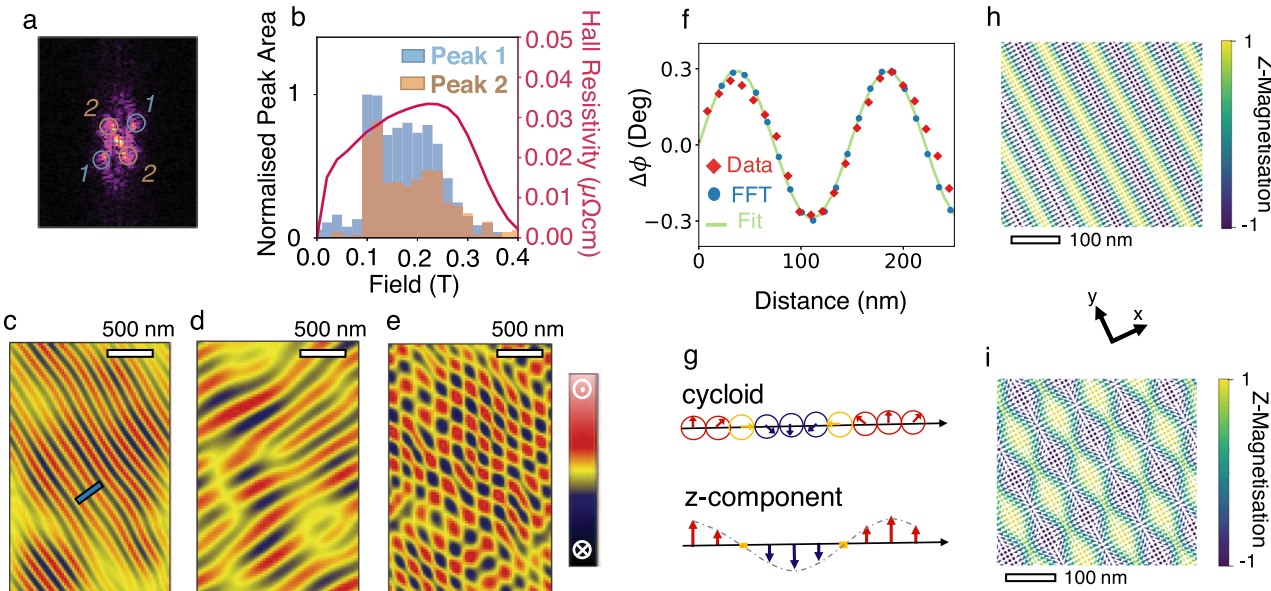

**Fig. 3 Fourier analysis of MFM images. a** Typical Fourier transform (here from 0.22 T), indicating the satellite peaks present during the periodic magnetic phase, with pairs of peaks labelled. **b** Normalised integration of the Fourier peak areas for the satellite peaks, plotted at each field, with the modulus of the Hall resistivity plotted for direct comparison. **c**, **d** MFM signals extracted by isolating the relevant respective peaks from the Fourier transform acquired from the 0.22 T scan. Line scan marked with a blue bar in **c** is plotted in **f**. **e** MFM signals extracted by isolating signals responsible for both peaks (convolution of **c** and **d**), resulting rectangular lattice is rotated ~28° to the scan direction. **f** Line scans extracted from raw data in Fig. 2b, and FFT extracted data in **c** fitted with a sine curve. **g** Schematic of a cycloidal phase, and how the z component of such a phase appears to vary. **h** 3D magnetisation simulation of a cycloidal phase with a single propagation vector. **i** 3D magnetisation simulation of an anisotropic incommensurate (IC) spin crystal, a cycloidal phase with two in plane perpendicular propagation vectors.

not topographic steps. This lack of step like features in saturating field has not always been replicated in previous studies[10,11].

A second phase, displaying sharper contrast (Fig. 2b), occurs at 0.1 T. The initial long stripes faintly visible before can be seen far more prominently (with a higher phase shift in the MFM signal). In order to analyse and quantify the periodicity in the real space, we perform FFT of the MFM images. FFT images were analysed revealing two pairs of peaks, a wider pair (peak 1, labelled in Fig. 3a), equivalent to a ~6.5 $\mu m^{-1}$ periodicity and the second pair (peak 2) slightly closer to the centre and 90° rotated to peak 1 resulting from a 4.5 $\mu m^{-1}$ periodicity. In order to quantify the Fourier images more rigorously, the peaks responsible for these periodicities, indicated in Fig. 3a were integrated at each field and are presented in Fig. 3b, alongside a modulus of the topological-like Hall resistivity response for comparison. Here, the integral bar plots reflect the increased contrast with a five times increment in relative peak area, and displays the onset of the additional periodicity perpendicular to the pre-existing periodicity. The onset of this additional periodic order is solely caused by the application of field. The ordering is clearly visible in the MFM images, where the stripes are broken up with small round evenly spaced domains forming on the stripes.

In Fig. 3b there is clear correlation between the emergence of the periodic ordering and the THE response with the periodic magnetic domain ordering appearing from 0 to 0.22 T, which is at the tip of the raised, topological-like peak in the Hall data. At higher fields as the THE signal decreases, domains take on a more random form (Fig. 2b) and with each successive increment of field the upward domains grow, merging into the saturated state. The FFTs indicate a vanishing of periodic order in this higher field range is, where the two pairs of satellite peaks disperse, leaving a growing central peak, indicating a distinct lack of periodicity in the domains. This final stage is expected of a system undergoing a magnetic switching process and this can be clearly seen as domain walls slowly move with each field increment as the rectangular lattice of bubble-like domains disperse and the green upward domains grow, reaching a saturated state.

To probe the symmetry breaking at the SRO-ferroelectric interface, STEM measurements were conducted. To image both heavy and light elements, annular bright field (ABF) imaging was used, as displayed in Fig. 4a with an atomic model overlayed. STEM analysis revealed the PTO to be in a tetragonal, mono-domain state, with a polarisation pointing uniformly out of the (100) growth plane. Quantified atomic displacement data acquired from the ABF show a decay in displacements, Fig. 4b, with the A site (Pb) to octahedral cage shift initially flat, before being shown to diminish upon approach of the PTO/SRO interface, as expected. However, the polarisation does not annihilate at the metal–ferroelectric interface, but continues across the interface with atomic displacements also present in the SRO, similar to the effect observed in the PTO/LSMO system[21]. The polarisation induced strain gradient in the SRO layer does not protrude the entire film, with around a third of the film in a more expected lattice. Whilst two stacked layers of differing SRO could give rise to a two-channel anomalous Hall effect as previously reported[5,10], no evidence of this two-stage switching is seen. We also note that the inhomogeneity in the film is different to that seen previously on two-channel bare SRO–STO films where the thickness of the film itself is inhomogeneous, as opposed to two effectively stacked layers with distorted unit cell. Whilst it is considered unlikely that a second switching regime is occurring in the ferroelectrically interfaced SRO, we also cannot rule out a certain contribution, as we can only image the top magnetic structure.

## Discussion

As shown in Fig. 2, after full saturation with −2 T applied field, at zero applied field, a remnant periodic domain pattern forms. The domain pattern evolves with the applied field into a more complex form with a rectangular lattice ordering, a pattern that is more obvious by closely analysing FFT of the MFM images. We start analysing the domain structure and its evolution with applied field by considering the domains characterised by the first emergent periodicity (labelled as peak 1 in Fig. 3a), before approaching the additional and simultaneous second perpendicular periodicities (labelled as peak 2 in Fig. 3a), and discuss the domains responsible for these orderings.

Isolating the signals responsible for the initial periodicity (peak 1) provides the filtered MFM image in Fig. 3c. Whilst initially such domains appear as stripes domains, examination of a line scan running perpendicular to these stripes (Fig. 3f) shows a sinusoidal oscillation in both the raw data, as well as the FFT filtered image (line scan positions marked in Figs. 2b and 3c, respectively). Classical stripe domains would not only be expected (especially across these length scales) to go through a period of transition (the domain wall) but also find a stable state, where the phase shift of the cantilever remains constant over a domain. Such an oscillation in the phase of the cantilever is consistent with a cycloidal phase, characterised by its amplitude of oscillation $C_x$ and the direction of 'stripe' propagation's vector, $q_x$ (lying

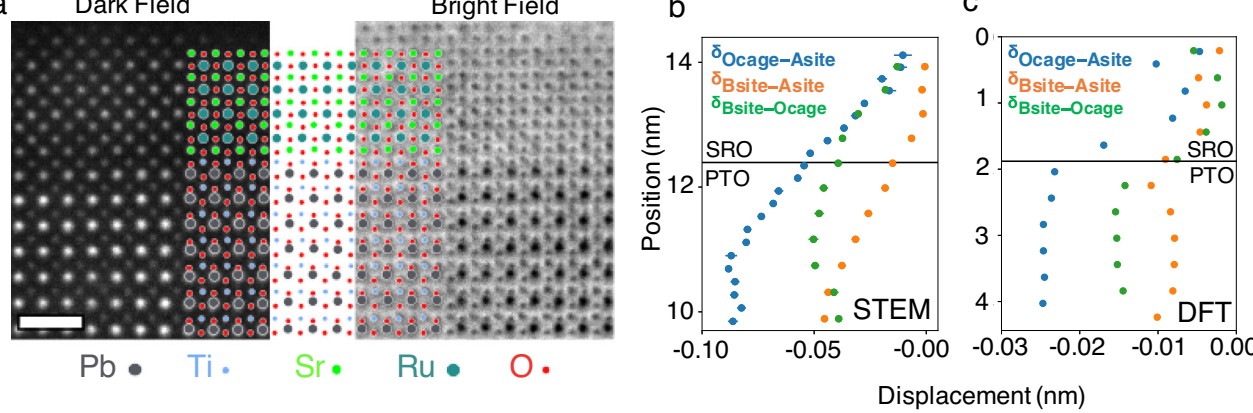

**Fig. 4 TEM and DFT results of the interface. a** Dark and bright field TEM images of the SRO–PTO interface with fitted atomic model overlayed. Scale bar is 1 nm. **b** Atomic displacements for the oxygen—A site shift (blue), the B site—A site shift (orange) and the B site to octahedral cage (green) acquired from the ABF images. **c** DFT simulated of the same atomic displacements around the SRO-PTO interface.

perpendicular to the stripes). A schematic of a cycloidal phase and how the MFM sensitive $z$ component varies spatially is presented in Fig. 3g, and an equation of the local magnetisation is equivalent to the first term in Eq. (1).

The sudden onset of stripes of bubbles in the raw images is mirrored in the Fourier analysis by the large and sudden increase to the maximum of both peaks bar areas at 0.1 T in the integral plot in Fig. 3b. This increase also marks the appearance of a second symmetry in the FFT (labelled as peak 2 in Fig. 3a, perpendicular to peak 1). This field induced transition is a phase transition from a cycloidal magnetic phase to a new emergent domain pattern settling to a steady state despite further field application, characterised in the FFT analysis with one periodicity having a higher intensity than the other. The new domains responsible for this new set of peaks (peak 2) are extracted and shown in Fig. 3d. Wider stripes are present in the extraction, and show slightly less rigorous order than the initial cycloidal phase (the periodicity of which is still present) and also express an oscillatory magnetisation, though with their own characteristic amplitude $C_y$ and propagation vector $q_y$ indicating a second, simultaneous cycloidal phase. The simultaneous existence of a two cycloidal domain patterns is described best as a 'double-q' vector state. Whilst the extraction of the pairs of peaks separately allows for the identification of two interwound cycloids, the manifestation of both domains can be seen best in the convolution of both periodicities, extracting both simultaneously, with the double-q cycloidal domains in Fig. 3e, presenting as an ordered bubble-like rectangular lattice.

Whilst this spin structure seemingly approximates a square skyrmion phase[23,24], precise and careful analysis shows otherwise. The topological nature of a skyrmion means that the local spin configuration has to be directly mapped onto a unit sphere of magnetisation. This is not the case observed here as the MFM response shows that the surrounding bubble-like features are not embedded in magnetisation pointing in the opposite direction but rather have tails stringing the regions together. These tails can be seen to join the bubbles meaning that there is not a full wrapping onto the unit magnetisation sphere and thus are not topological objects. Indeed, their interaction in rows is evidence for a lack of topological isolation required of a stable skyrmion state. The negation of a skyrmion phase combined with strong evidence of two interwound cycloidal phases presenting as the observed rectangular lattice of bubble-like features are more likely associated with an anisotropic double-q state[23,25], often referred to as an incommensurate (IC) spin crystal.

This IC phase takes the form of a linear sum of two cycloids with perpendicular propagation directions. The local magnetisation is expressed in Eq. (1) for two cycloids where one is propagating along the $x$ with a modulation $q_x$ and the other along $y$ with $q_y$. The anisotropic nature means the amplitude of these cycloids are not equal ($c_x \neq c_y$). The equation for the purely cycloidal phase is just the first term of the following equation:

$$m(x,y) = I_n(x,y)\left(c_x\begin{bmatrix} \sin q_x x \\ 0 \\ \cos q_x x \end{bmatrix} + c_y\begin{bmatrix} 0 \\ \cos q_y y \\ \sin q_y y \end{bmatrix}\right) \quad (1)$$

$$I_n(x,y) = \frac{1}{\left((c_x \sin q_x x)^2 + (c_y \cos q_y y)^2 + (c_x \cos q_x x + c_y \sin q_y y)^2\right)} \quad (2)$$

Here, the spin texture has been normalised with $I_n(x,y)$ (defined in Eq. (2)), which was introduced to ensure all magnetic moments are the same length[23,24]. Figure 3h, i shows the simulation of the magnetisation of the cycloidal phase and the

anisotropic double-q structure, respectively. Values for $q_x$ and $q_y$ ($|q_x| = 153$ nm, propagating 28° to the horizontal and $|q_y| = 222$ nm propagating 118° from the horizontal) were obtained from Fourier peak positions/rotations of the MFM response, while the ratio of $c_x$ and $c_y$ $\left(\frac{c_y}{c_x} = 0.7\right)$ was determined from the relative difference in Fourier peak intensity.

Double-q vector magnetic phases have been subject to frequent investigation, both theoretically[23,24] and experimentally[25–27,29,30]; however, any identification of such regions has been through reciprocal space techniques such as neutron diffraction, and associated with ac-magnetic susceptibility measured phase diagrams. The results obtained from MFM in this study closely agree with this magnetic spin texture, with this report being the first to display real-space images of such spin crystals. Fourier analysis of the MFM exhibits similar behaviour to neutron diffraction data of this phase[25,27,29], specifically the separation of peaks and differences in intensity, adding further evidence to this characterisation, whilst still providing the raw images and information that most reciprocal space techniques lack.

When considering the impact that a spin crystal such as the one proposed would have on the transverse Hall resistivity, it helps to introduce the local scalar spin chirality $\chi$, defined as

$$\chi = \mathbf{S_i} \cdot (\mathbf{S_j} \times \mathbf{S_k}) \quad (3)$$

where $\mathbf{S_i}$, $\mathbf{S_j}$, and $\mathbf{S_k}$ refer to three neighbouring spins in the magnetic spin texture[31–38]. Whilst explicit calculation of a resistivity from a spin texture is analytically challenging, the dependence of a topological-like Hall response on $\chi$ is widely accepted[1,31,33,35,38–41]. It is noted that whilst the net chirality when summed across the magnetic unit cell can be zero, critical for an additional, topological Hall contribution is for the local $\chi$ to be non-zero[1,31]. Local scalar spin chirality causes a Berry curvature in momentum space (and also in the real space for topologically non-trivial textures such as skyrmions)[1,36,41], which can be understood as a fictitious magnetic field, with the Berry phase acquired by conduction electrons as the angle subtended by the three spins in Eq. (3).

For the double-q cycloidal magnetic spin texture in this investigation, a non-zero $\chi$ has already been theoretically established[23,24]; however, a non-zero $\chi$ can be directly identified from observation of the spin texture in Fig. 3i. For any three neighbouring spins across the smoothly varying non-coplanar magnetisation texture[35,38], the angle subtending the three spins is clearly non-zero, which is to say that $\chi$ is non-zero.

The chiral nature of the magnetism in this model also offers an alternative explanation for the origin of the topological-like Hall response[23,26,42–44] with the real-space spirals sufficient to cause conduction electrons to accumulate a Berry Phase, thus providing an additional contribution to the Hall effect. Hysteretic behaviour of the Hall effect corroborates magnetic nature of its origin from the anisotropic double-q structure. The magnetic phases are lost with application of a high magnetic field polarising the state and only reformed with the removal of the saturating field.

Cycloidal domains have been previously shown to be caused with an iDMI[45–47] and indeed the emergence of such double-q states (both cycloidal and others) are associated with the iDMI, caused by broken surface inversion symmetry at the thin film interface. This is visible when considering the B site shift relative to the A site, as well as the shift from the A and B sites from the octahedral cage. Surface symmetry breaking of this nature is critical for the iDMI, and has been associated with potential skyrmion behaviour, inducing a THE in other SRO-ferroelectric systems[5,6], though the displacements measured here are much larger than those previously reported.

Density functional theory (DFT) calculations were conducted to further verify the previously reported mechanism (Fig. 4c). DFT revealed a very similar behaviour to that observed in the electron microscope. The same shifts behaviours were found in the DFT, with the greatest displacement found to be from the A site in the octahedral cage. The lack of depolarisation in the PTO when approaching the interface is due to the lower thickness of PTO used in the calculation. Calculations were also performed taking into account the octahedral rotation, however this had a minimal effect on the induced polarity.

In summary, the data presented above show that the strong polarisation present in the PTO ferroelectric layer induces a decaying polarisation into the top SRO layer over 3–4 unit cells breaking the surface inversion symmetry and generating a strong iDMI. As the magnitude of an iDMI is increased, the energy cost of domain walls is reduced to below zero, instead favouring a chiral, Néel winding of domains (hence the association of the iDMI and the emergence of stable domain structures such as magnetic skyrmions). Instead of skyrmions, the domains that emerge are an IC spin crystal, which is seen to coincide with a significant topological-like Hall effect in this SRO/PTO system, leading to the possibility that this is caused by the magnetic structure due to its intrinsic chirality.

Such domains have yet to be found in any other SRO DMI systems (and indeed yet to be imaged in real space in any material); however, the presence of PTO as the symmetry breaker provides a sufficiently large DMI interaction to stabilise a zero-field cycloidal and field stabilised IC spin crystal phase. There is still much to be understood about the iDMI-mediated magneto-electric coupling from both an experimental and theoretical viewpoint.

## Methods

**Sample growth and electrical measurements**. Two samples comprising SRO(6 u. c.)–PbTiO$_3$(20 u.c.) bilayers were simultaneously grown on vicinal SrTiO$_3$(001) substrates (CrysTec GmbH) by pulsed laser deposition using a 248 nm KrF excimer laser. The substrates were BHF etched prior to deposition in buffered HF and then annealed at 1100 °C for 1 h in air to achieve vicinal TiO$_2$ surface termination. The deposition conditions for both the SRO and the PTO thin films were set to ~1 J/cm$^2$ laser fluence, 3 Hz ablation frequency, and 0.1 mbar partial O$_2$ chamber pressure. The substrate temperature was 650 and 600 °C for SRO and PTO, respectively, achieving step flow growth. One sample was used for the electrical measurements, which were performed on 25 μm × 480 μm Hall bar devices, prepared by photolithography, with e-beam deposited Au–Ti contacts.

**Magnetic force microscopy**. The other, sister sample was used for MFM images, which were acquired in a dual pass mode. The first pass is a normal non-contact mode in order to attain topographic contributions, which is then used to remove topographic interference on the second pass, a ~100 nm lifted pass of the same line scan, now detecting only the magnetic contributions. A low temperature AFM (Attocube attoLiquid 2000) provided with interferrometric SPM system and 9 T axial/1 T × 1 T vector magnetic field capability and 1.8–300 K temperature range. Nanosenors PPP-MFMR cobalt coated tips, ~80 KHz resonance, were used. Gwyddion[48] image processing package was used to align rows. Fourier analysis was performed in Python, with a 5 × 5 pixel window, at every field, thus providing the field dependent bar plots in Fig. 3b. Plotted above them is the modulus of the 'trough' like region in the Hall resistivity for comparison.

**STEM**. Atomic-resolution STEM images were acquired using a double aberration-corrected JEOL ARM-200F operating at 200 kV. Annular dark field images were formed using a collection angle of 45−180 mrad, and ABF images were formed using collection angles of 11−23 mrad[49]. To eliminate scan distortions and drift 16 images were acquired with orthogonal scan directions and reconstructed into a single image using the SmartAlign routine[50]. Atomic coordinates were measured by finding local pixel maxima which were then refined using non-linear least squares fitting, taking into account nearest neighbour peak contributions. From this all relative displacements can be measured. TEM specimens were prepared using a Tescan AMBER focussed ion beam microscope using standard lift-out procedures.

**DFT**. The DFT simulations were performed using Quantum Espresso kit within the generalized gradient approximation[51]. The exchange–correlation functionals used are given in Perdew–Burke–Ernzerhof parametrization[52,53]. We used Vanderbilt ultrasoft pseudopotentials[54] with kinetic energy cut-off equal to 75 and 450 Ry for the charge density cut-off. The valence states of the elements used are Pb 5d10 6s2 6p2, Sr 4s2 4p6 4d1 5s1 5p0, Ti 3s2 3p6 4s2 3d1, Ru 4d7 5s1 5p0, and O 2s2 2p4. Brillouin zone integration was performed on an automatically generated Monkhorst–Pack k-mesh[55] with a Gaussian smearing of 0.01 eV. Total force per atom is converged to less than 30 meV/Å. The supercell consists of 6.5 u.c. of SRO and 5.5 u.c. of PTO along Z axis and one unit cell of 3.905 Å (=STO) in the XY plane. The system is designed with asymmetric interfaces, PbO–RuO$_2$ and TiO$_2$–SrO, and octahedral tilts are not considered.

## Data availability

The data that support the findings of this study are available from the corresponding author upon reasonable request.

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

## Acknowledgements

The financial support by the Engineering and Physical Science Research Council (EPSRC) for the PhD studentships and related grants EP/M022706/1, EP/M028186/1, EP/P031544/1, and EP/P025803/1 is gratefully acknowledged. M.A. acknowledges the Theo Murphy Blue-sky Award of Royal Society. S.D.S. and J.J.P.P. thank Steven Hindmarsh for his expertise and help with Focussed Ion Beam TEM sample preparation. S.D.S and S.J.R.H. thank Juba Bouaziz for his discussion on the surrounding theory and S.D.S. thanks George Rowlands for his valued discussion.

## Author contributions

M.A conceived the idea, S.D.S. conducted and analysed the MFM measurements, D.E.D. prepared the samples and conducted the electrical measurements, S.J.R.H. performed the magnetisation simulations and contributed to the discussion, D.R. performed the DFT calculations, J.J.P.P. performed the TEM measurements and analysis, and A.M.S contributed to the discussion. All authors contributed to the manuscript writing.

## Competing interests

The authors declare no competing interests.
