## [Peer Review File · Nature Communications]

REVIEWER COMMENTS

Reviewer #1 (Remarks to the Author):

This manuscript presents an interesting anomalous Hall effect (AHE) study on SrRuO₃/PbTiO₃ bilayer films. The AHE data here show peak, and this feature is often interpreted as topological Hall effect (THE) that is commonly used to show the existence of the skyrmions. However, the THE-like behavior has been observed in many systems by many groups and there is no consensus on its origin the main findings of the current work are that field dependent Magnetic Force Microscopy measurements throughout the key fields where the 'THE' presents, revealing the emergence to two periodic, chiral spin textures. While the results are important in improving our understanding of the origin of the THE-like features. In my view, this is an interesting work and I would recommend it for further consideration as a regular article.

However, the following arguments are not entirely clear and conclusive at its present version.

1. The present temperature-dependence does only contain 10 K, 20 K and 40 K. The authors said that "the as-grown system shows emergent Topological-like Hall effect below 80 K" and "whereas the hysteresis is drastically diminished above 40 K". If possible, the authors should add more temperature points especially from 40 K to 100 K. This will provide a much better understanding of the evolution of the THE-like with temperature.
2. The authors found that the polarization does not annihilate at the metal-ferroelectric interface, continues across the interface with atomic displacements also present in the SRO by STEM and DFT calculations. According the authors' results, the SRO is inhomogeneous. In the work by Ref 10 and Ref 11, they reported the inhomogeneity SRO showing the coexistence of two channels with opposite Berry curvature, leading the THE-like features. The authors should explain that reason.
3. Fig. 3c-e is the key point of the paper. "The first real-time space Observation of such chiral spin Crystal Skyrocket". This point need more detailed description of its connection to MFM data.
4. Page 9: The author mentioned "the phase shift of the cantilever remains unchanged." Please provide further explanation for the reason.
5. Page 12: The authors mentioned that when approaching the interface, the lack of depolarization field in the PTO is due to the lower thickness of PTO used in the DFT calculation. However, in this calculation, the suitable thickness of the PTO was suggested to be used.
6. The authors claim the domains seen to emerge is an incommensurate (I-C) spin crystal which, due to its intrinsic chirality, generates a significant topological Hall effect in this SRO/PTO system. However, in ref 10, they reported the inhomogeneity domain in SRO, leading the THE-like features. How to understand the difference between the inhomogeneity domain in SRO leading two channel AHE and domain of incommensurate (I-C) spin crystal leading to the THE?
7. The author mentioned that "These tails can be seenof a stable skyrmion state. ". Are there any other evidences to confirm the reliability of the method, which tells the difference between skyrmions and incommensurate (I-C) spin crystal through Fourier transform and convolution process MFM data?
8. I suggest to correct some inconsistencies in the manuscript: The authors use a mixture of "polarization" and "polarization" in their manuscript. I suggest to stick to the "polarization" or "polarization" throughout the manuscript.

Reviewer #2 (Remarks to the Author):

In the manuscript "Real-space Observation of Ferroelectrically Induced Magnetic Spin Crystal in SrRuO₃" by Seddon et al., the authors report on the topological Hall effect (THE) and Magnetic Force Microscopy (MFM) in SrRuO₃-PbTiO₃. The observed chiral spin textures correlates well with the THE. The inversion symmetry breaking is observed by Scanning Tunnelling Electron Microscopy (STEM),

suggesting that the an interfacial Dzyaloshinskii–Moriya interaction (IDMI) is a key of the system.

I found that the results shown in the manuscript are interesting. The real space observation of the spin texture may provide us to solve the controversy in the THE-like behaviors in the SRO-related systems. I will recommend publication after the following points are addressed:

1. The authors claim that the THE is driven by the chiral spin structure observed by MFM, while no quantitative discussion is given. The authors should comment on it.

2. I do not fully understand a message of Fig. 4. The data indicates that there is a finite degree of asymmetry in the SRO layers, which has been already reported in Ref. 5 and 6. Is this the only claim of Fig. 4? The authors mentioned that the atomic displacements are much larger than those in Refs. 5 and 6. Then we can naturally expect that the IDMI and the resultant THE are much larger as well; is it really the case?

3 (minor)

Page 4: "Therefore, the THE is the main contribution in the Hall response above the high-field saturation value"

I am a bit confused; this "above" should be "below", shouldn't it?

Reviewer #3 (Remarks to the Author):

The authors declared that they revealing the emergence to two periodic, chiral spin textures by the a topological Hall effect based on the field dependent Magnetic Force Microscopy (MFM) measurements. The surface inversion symmetry breaking and the DM interaction role in the interface is confirmed and discussed by the observation of Scanning Tunnelling Electron Microscopy, and also calculated by Density Functional Theory. I believe that the work is in the front, and the work extend the SRO properties and it application, may be. Details experiments and mechanism have been supplied. English is well organized. Based on the above, I recommend the paper to be published in NC in the present.

Response to Reviewer #1

Question 1.1 *The present temperature-dependence does only contain 10 K, 20 K and 40 K. The authors said that “the as-grown system shows emergent Topological-like Hall effect below 80 K” and “whereas the hysteresis is drastically diminished above 40 K”. If possible, the authors should add more temperature points especially from 40 K to 100*

Answer 1.1) The extra temperatures of Hall resistivity are now presented in Fig. 1. Adding all the main temperatures to a single figure would have been confusing to the eye, instead we have changed the plot to a stack plot, where the emergence of the feature is clear and can be easily compared between the temperatures.

Action 1.1) We have included of Hall resistivity acquired at all temperature to show the Hall resistivity both with and in the absence of a topological Hall effect. We have also changed the colour scheme of the other plots to ensure 40k is consistently the right colour throughout, now that figure 1 was adapted.

Question 1.2) *The authors found that the polarization does not annihilate at the metal-ferroelectric interface, continues across the interface with atomic displacements also present in the SRO by STEM and DFT calculations. According the authors’ results, the SRO is inhomogeneous. In the work by Ref 10 and Ref 11, they reported the inhomogeneity SRO showing the coexistence of two channels with opposite Berry curvature, leading the THE-like features. The authors should explain that reason.*

Answer 1.2) Here the point in question requires a direct consideration of the nature of described inhomogeneity. In Ref 10 and Ref 11 the inhomogeneity which gave rise to a two channel anomalous Hall effect in bare SRO-STO (no PTO) is physical, i.e. one unit cell difference in thickness that arises due to slight step terrace overstacking, and directly linked to substrate step terraces. The inhomogeneous nature of the SRO in our film arises from a unit cell deformation, due to the induced polarisation from the PTO layer in the bottom 3 u. c. of the SRO. In this sense, the

35 nature of inhomogeneity is at a more fundamental level than a simple variation in
thickness or local stoichiometry. The broken inversion symmetry caused by the
ferroelectric polarisation is what gives rise to the interfacial DMI responsible for the
stability of the spin texture.

**Action 1.2)** The nature of the inhomogeneity has been clarified in the main text.

The following sentence has been added on page 8 last paragraph before Discussion:

“Such inhomogeneity in the film is different to that seen previously on SRO-STO
films^{5,10} where the thickness of the film itself is inhomogeneous.”

**Question 1.3)** *Fig. 3c-e is the key point of the paper. “The first real-time space
Observation of such chiral spin Crystal Skyrocket”. This point need more detailed
description of its connection to MFM data.*

**Answer 1.3)** We realise we did not emphasise that all previous experimental
observations have been from reciprocal space measurement regimes (such as
neutron diffraction), unlike MFM which provides both real and reciprocal images.

**Action 1.3)** A sentence beginning “The results obtained by MFM...” has been
reworded in paragraph two of page 11, to add emphasis to the MFM detection of this
spin crystal

**Question 1.4)** *Page 9: The author mentioned “the phase shift of the cantilever
remains unchanged.” Please provide further explanation for the reason.*

**Answer 1.4)** This sentence referred the phase shift of a cantilever remains at a
constant, horizontal level when scanning over a single stripe domain before a large
change at the domain wall between a second stripe domain, where the cantilever
returns to a steady state again. This is as opposed to the cycloidal domains that
sinusoidally oscillate.

**Action 1.4)** The following sentence on page 9 has been adapted to avoid confusion:

“... where the phase shift of the cantilever remains constant whilst over a domain.

Such an oscillation in the phase of the cantilever is consistent with a cycloidal phase

...”

**Question 1.5)** *Page 12: The authors mentioned that when approaching the*
*interface, the lack of depolarization field in the PTO is due to the lower thickness of*
*PTO used in the DFT calculation. However, in this calculation, the suitable thickness*
*of the PTO was suggested to be used.*

**Answer 1.5)** The sample grown had a PTO layer of 20 unit cells (u.c.) unlike the
simulation which was limited to 5 u.c. both for the reasons described in the text (i.e.
depolarising field) and also due to computational cost of simulating 15 u.c. more.

**Action 1.5)** No action taken.

**Question 1.6)** *The authors claim the domains seen to emerge is an incommensurate*
*(I-C) spin crystal which, due to its intrinsic chirality, generates a significant*
*topological Hall effect in this SRO/PTO system. However, in ref 10, they reported the*
*inhomogeneity domain in SRO, leading the THE-like features. How to understand*
*the difference between the inhomogeneity domain in SRO leading two channel AHE*
*and domain of incommensurate (I-C) spin crystal leading to the THE?*

**Answer 1.6)** This difference in inhomogeneity has already been addressed in our
reply to comment 2). In terms of understanding the difference between the I-C spin
crystal found here, and the previous domains reported in bare SRO/STO system, the
simplest is perhaps to consider how they react to applied field. In Ref 9,10 and
actually in 12, the domains reported are the result of magnetic switching from one
field polarised state to another, ie the system transitioning from down to up. With
each successive increment of field, their domains increase in size until they join and
merge into the opposite field polarised state. A critical point here is their domains are
not periodic. In contrast, the spin crystals stabilised the SRO-PTO-STO system
studied here do not grow and change like this, instead they form a steady state,
which from zero field is a cycloidal *phase*. These cycloids, which appear without
careful inspection simply as stripes are shown to change into a separate (I-C) spin
crystal on the application of further magnetic field. The system here is undergoing a
field induced magnetic phase transition (much like a system originally starting in say
a helical phase, and with the application of field attaining a skyrmion phase). The

second spin crystal similarly is stabilised in successive field increments, before
slowly dispersing into more randomly formed domains, as it then grows to field
polarised. This final phase, labelled ferromagnet, is very similar to the switching seen
in the two channel hall effect papers. In summary, we have two field stabilised,
periodic domain patterns, whereas they have different regions that switch at different
fields due to slight differences in film thickness due to step terraces.

**Action 1.6)** No action taken.

**Question 1.7)** *The author mentioned that "These tails can be seenof a stable*
*skyrmion state. ". Are there any other evidences to confirm the reliability of the*
*method, which tells the difference between skyrmions and incommensurate (I-C)*
*spin crystal through Fourier transform and convolution process MFM data?*

**Answer 1.7)**

MFM was the major provider of evidence when identifying the measured spin
texture. MFM as a diagnostic tool for spin textures of this scale has already been
firmly established in the literature [see Milde *et al.*, Science, 340, 2013,
doi: 10.1126/science.1234657 or Zhang *et al.*, Nano Letters, 16,
2016, 10.1021/acs.nanolett.6b00845], especially when identifying and studying
Skyrmion lattices. The spin crystal periodicity is fairly large relative to the tip radius,
and furthermore, the key concern – the tails between bubble-like domains – occur off
scan axis and in only one direction. Indeed the direction the tails span between
bubbles is that of perpendicular to the dominant periodicity, parallel to the sub-
dominant periodicity. This lack of complete ‘inversion’ between domains indicates a
lack of topological protection, and results in the Fourier Transform peaks having
different intensities. Were the spin texture to be the unlikely square Skyrmion state,
bubbles would be expected to be entirely isolated, and their Fourier transform
expected to be four equally intense points reflecting this (much like the six-fold
symmetries seen in the above works by Milde and Zhang). This mismatch in intensity
for the Fourier peaks is reflected in the previous experimental detections of this IC
spin crystal, crucially in reciprocal space detection techniques, such as Neutron
Diffraction.

**Action 1.7)** No action taken

**Question 1.8)** *I suggest to correct some inconsistencies in the manuscript: The*
*authors use a mixture of “polarization” and “polarization” in their manuscript. I*
*suggest to stick to the “polarization” or “polarization” throughout the manuscript.*

**Action 1.8)** We have changed all to the correct English spelling: “polarisation”. There
was one occurrence on page 7.

**Reviewer #2**

**Question 2.1)** *The authors claim that the THE is driven by the chiral spin structure*
*observed by MFM, while no quantitative discussion is given. The authors should*
*comment on it.*

**Answer 2.1)** Whilst the emergence of additional Hall resistivity contributions due to a
real space Berry curvature are well understood, there currently exists no theoretical
work comparing the magnitude the induced contribution due to this spin texture, with
calculations focussing simply on the Hall Conductivity (/resistivity) as a bulk property.
A detailed calculation of a such properties for an ultra-thin film go beyond the scope
of this work.

Anyway, we would like to refrain ourselves in speculating and making a quantitative
analysis based solely on a non-quantitative measurement such as MFM.

We parenthetically mention that our data presented in Fig. 1 showing topological Hall
resistivity is raw Hall resistivity data, compared to most of the data presented in the
literature that is process data by the so-called anti-symmetrisation aiming to subtract
the anomalous and linear Hall contribution.

**Action 2.1)** See action 2.2.

**Question 2.2)** *I do not fully understand a message of Fig. 4. The data indicates that*
*there is a finite degree of asymmetry in the SRO layers, which has been already*
*reported in Ref. 5 and 6. Is this the only claim of Fig. 4? The authors mentioned that*

*the atomic displacements are much larger than those in Refs. 5 and 6. Then we can*
*naturally expect that the IDMI and the resultant THE are much larger as well; is it*
*really the case?*

**Answer 2.2)** Common among the two other reports (Refs. 5,6) of combining SRO
with a ferroelectric layer is the emergence of a THE (-like) response. When
combined with BiFeO₃ in 5, the SRO layer was around 20 nm, so the magnitude of
the effect is not possible to compare. In the case of SRO combination with BaTiO₃,
whilst the SRO film thickness are more comparable (although with the 4 u.c. they
used, that would imply only 1 u.c. of 'nomal' SRO that isn't polarised by the BaTiO₃),
there is still much to be understood about the dominating effects at the interface of a
ferroelectric and ferromagnet. Whilst the polarisations (displacements) measured in
our system were bigger, the protrusion into the sample were both around the same
(3 u.c.). More theoretical work is required on how the ferroelectric layer induces the
interfacial DMI.

**Action 2.2)** A sentence has been added to the end of the conclusion, beginning with
"... there is still much" and discussing the lack of understanding of the dominate
properties of the interface that cause the iDMI have been added for clarity, and goes
some length to address 2.1) in the main text as well.

**Question 2.3)** *Page 4: "Therefore, the THE is the main contribution in the Hall*
*response above the high-field saturation value" I am a bit confused; this "above"*
*should be "below", shouldn't it?*

**Answer 2.3)** This discrepancy arises from the convention of how the Hall Hysteresis
are plotted, combined with the fact that we plot a modulus in Fig 3.

**Action 2.3)** This has been corrected in the main text, on page 4, paragraph one,
sentence beginning "Therefore, the THE..."

REVIEWER COMMENTS

Reviewer #1 (Remarks to the Author):

The authors have replied all questions arisen from previous comments. This work is an interesting work and then I recommend it is published as a regular article after the following minor revision.

For question 2, the author said "The inhomogeneous nature of the SRO in our film arises from a unit cell deformation". I agree with it basically. However, the deformations of the film are different from bottom to top u.c. of the SRO from STEM. Thus, the authors supposed the different deformation can induce different magnetic and electric properties, leading to two channels AHE and THE-like features. The authors are better to make further discussions on that.

Reviewer #2 (Remarks to the Author):

I have read through the response. It is quite unfortunate that the authors hardly consider my comments, in particular, Question 2.1. Since I am afraid that the term "quantitative" was somewhat misleading, let me rephrase my original comment. The authors demonstrated that the correlation between THE-like transport behavior and their MFM images. Then they attribute the former to the latter; this is simply a jump in logic. As far as I understand, the authors discuss this issue in the short paragraph (ll. 226-231) but it only claims that some chiral magnetism may affect Hall effect in general. I do not think that it is sufficient. There should be some theoretical papers corresponding to "double-q" systems, if not exactly the same as the current experimental situation; some sort of semi-quantitative discussion is possible. Otherwise I do not understand why the authors believe that the observed magnetic pattern results in the THE-like behavior. While I never expect perfect coincidence between the theory and the experiments, the above mentioned correlation is quite weak as evidence.

**Response to Reviewer #1**

**Question:** *For question 2, the author said “The inhomogeneous nature of the SRO in our*
*film arises from a unit cell deformation”. I agree with it basically. However, the*
*deformations of the film are different from bottom to top u.c. of the SRO from STEM. Thus,*
*the authors supposed the different deformation can induce different magnetic and electric*
*properties, leading to two channels AHE and THE-like features. The authors are better to*
*make further discussions on that.*

Answer: We agree that treatment of the inhomogeneity of the film needs to be handled
delicately. Particularly as, as you say, we have effectively two layers of SRO, one under an
immense strain gradient induced by the Ferroelectric, and the other, the top ~third of the film
in an expected lattice for strained SRO/STO. As the two channel Hall effect reported in bare
SRO/STO is effectively a result of different film thicknesses switching at different times, we
consider it unlikely that an out of plane field, acting across the two ‘layers’ is likely to
manifest itself, nevertheless we have reworded the discussion to highlight the presence of this
inhomogeneity, and whilst no experimental evidence in the top layer of two channel
switching is present, we cannot rule it out either.

Action: Rewording/ addition to existing paragraph on page 8, sentence beginning “**The**
**polarisation induced strain gradient...**”

**Reviewer #2**

**Question:** *I have read through the response. It is quite unfortunate that the authors hardly*
*condider my comments, in particular, Question 2.1. Since I am afraid that the term*
*"quantitative" was somewhat misleading, let me rephrase my original comment. The authors*
*demonstrated that the correlation between THE-like transport behavior and their MFM*
*images. Then they attribute the former to the latter; this is simply a jump in logic. As far as I*
*understand, the authors discuss this issue in the short paragraph (ll. 226-231) but it only*
*claims that some chiral magnetism may affect Hall effect in general. I do not think that it is*
*sufficient. There should be some theoretical papers corresponding to "double-q" systems, if*

*not exactly the same as the current experimental situation; some sort of semi-quantitative*
*discussion is possible. Otherwise I do not understand why the authors believe that the*
*observed magnetic pattern results in the THE-like behavior. While I never expect perfect*
*coincidence between the theory and the experimemnts, the above mentioned correlation is*
*quite weak as evidence.*

**Answer:** It was never our intention to give the impression of not considering the Reviewer's
comments. Indeed contrary to that, the Reviewer's discussion points, especially those relating
to linking atomic displacements in the SRO to a quantified DMI re-ignited several
discussions on ways to quantify and compare the displacements to the resultant Hall peak.

The same issue arises when attempting to directly quantify a Hall resistivity response from a
magnetic spin texture. On the surface such a calculation seems fairly straightforward. Indeed,
in the case of the topologically protected Skyrmion, fairly back of the envelope approaches
can be employed [*Nature Materials* 17, 1087–1094 (2018), *Nature Communications* 9,
213 (2018)], relying on the real space Berry curvature caused by the chirality of the
skyrmion, and skyrmion density to compare expected Topological Hall effects. It should be
noted however that this approach does not consider the fact that a chiral spin texture also
causes a Berry curvature in the momentum space, a curvature that is independent of the
topological nature of the underlying spin texture. For a topologically trivial spin texture all
Berry curvature occurs in momentum space [Jour. Phys. Soc. Jap, 2004, Vol. 73, No. 10 : pp.
2624-2627]

The true challenge of the direct calculations the Reviewer propose are that to analytically
solve the effect of a magnetic spin texture for a single conduction electron, one must formally
begin with quantum mechanical interactions between conduction electrons and localised spin
electrons. Further complicating our particular case is that the spin texture is incommensurate,
which is to say more specifically that the magnetic unit cell is much larger than the structural
unit cell.

The current state of the art of the literature is that whilst it is widely acknowledged that a
'Topological Hall effect' arises when the scalar spin chirality (χ) is non zero. Many high
profile works [for references see action in main text] have relied on this relationship, and the

demonstration of a non-zero χ when associating magnetic spin textures with a Hall response.
This includes a work published in Nature Communications two weeks ago, where such
calculations would perhaps be even more relevant considering the topic of the work actually
considers specifically the magnitude of the Hall response they find [*Nature*
*Communications* volume 12, 317 (2021)]. Whilst several papers exist exploring the direct
dependencies [Phys. Rev. Lett. 124, 096602 2020 and Jour. Phys. Soc. Jap. paper above], the
application of these methods is far from trivial, and we consider such lengthy calculation well
beyond the scope of this work.

**Action:** In order to follow suit with the literature and to highlight the links between the scalar
spin chirality χ an extra two paragraphs in the main text have been added (page 11/12,
paragraph beginning “When considering the impact”). In these, we define χ , outlining with
the help of the literature its relevance and proportionality to the Hall effect, and reference two
specific papers which confirm the non-zero nature of χ in this particular spin crystal. We also
lead the reader through considering how χ must be non zero, when considering its definition
in relation to the spin texture reported. We hope this highlights more clearly the link between
the spin texture and Hall effect in an appropriately semi-quantitative way. We also amended a
sentence in the conclusion/summary on page 13 beginning “Instead of skyrmions” in order to
emphasise the role chirality plays.

REVIEWERS' COMMENTS

Reviewer #1 (Remarks to the Author):

The authors answered all of my concerns. The current version is acceptable for the publication.

Reviewer #2 (Remarks to the Author):

By reading the second response, I appreciate the notion of the authors. I believe that the significance of the manuscript is better demonstrated due to the modifications made by the authors; I recommend the current version for acceptance.